**Data Availability Statement:** Personal health data underlying the findings are protected by the French

# A new trajectory approach for investigating the association between an environmental or occupational exposure over lifetime and the risk of chronic disease: Application to smoking, asbestos, and lung cancer

Emilie Lévêque[1], Aude Lacourt[1], Viviane Philipps[1], Danièle Luce[2], Pascal Guénel[3], Isabelle Stücker[3], Cécile Proust-Lima[1], Karen Leffondré[1]*

1 Univ. Bordeaux, Inserm, Bordeaux Population Health Research Center, UMR 1219, Bordeaux, France, 2 Univ. Rennes, Inserm, EHESP, Irset (Institut de recherche en santé environnement et travail), UMR_S 1085, Pointe-à-Pitre, France, 3 Inserm, CESP, Univ. Paris-Saclay, UVSQ, Villejuif, France

* Karen.leffondre@u-bordeaux.fr

## Abstract

Quantifying the association between lifetime exposures and the risk of developing a chronic disease is a recurrent challenge in epidemiology. Individual exposure trajectories are often heterogeneous and studying their associations with the risk of disease is not straightforward. We propose to use a latent class mixed model (LCMM) to identify profiles (latent classes) of exposure trajectories and estimate their association with the risk of disease. The methodology is applied to study the association between lifetime trajectories of smoking or occupational exposure to asbestos and the risk of lung cancer in males of the ICARE population-based case-control study. Asbestos exposure was assessed using a job exposure matrix. The classes of exposure trajectories were identified using two separate LCMM for smoking and asbestos, and the association between the identified classes and the risk of lung cancer was estimated in a second stage using weighted logistic regression and all subjects. A total of 2026/2610 cases/controls had complete information on both smoking and asbestos exposure, including 1938/1837 cases/controls ever smokers, and 1417/1520 cases/controls ever exposed to asbestos. The LCMM identified four latent classes of smoking trajectories which had different risks of lung cancer, all much stronger than never smokers. The most frequent class had moderate constant intensity over lifetime while the three others had either long-term, distant or recent high intensity. The latter had the strongest risk of lung cancer. We identified five classes of asbestos exposure trajectories which all had higher risk of lung cancer compared to men never occupationally exposed to asbestos, whatever the dose and the timing of exposure. The proposed approach opens new perspectives for the analyses of dose-time-response relationships between protracted exposures and the risk of developing a chronic disease, by providing a complete picture of exposure history in terms of intensity, duration, and timing of exposure.

Data Protection Act and cannot be shared publicly. The large number of variables allows data to be indirectly identifiable, and making such data freely available is prohibited. Furthermore, an authorization from the CNIL (Commission nationale de l'informatique et des libertés), the French Data Protection Authority, may be required to transfer the data, especially abroad. Data from this study can be obtained upon request from the ICARE steering committee (contact@irset.org or http://cesp2018.vjf.inserm.fr/en/basic-page/how-find-us-contact), as well as from the corresponding author (karen.leffondre@u-bordeaux.fr).

**Funding:** The results reported herein correspond to specific aims of source of support [2013/1/177] to investigator KL from The French National Research Program for Environmental and Occupational Health of Anses with support of the Cancer TMOI of the French National Alliance for Life and Health Sciences (AVIESAN). EL was supported by a doctoral award from the French national institute of cancer (INCa). The funders had no role in study design, data collection and analysis, decision to publish, or preparation of the manuscript.

**Competing interests:** The authors have declared that no competing interests exist.

## Introduction

Occupational and environmental exposures often extend over lifetime, with a time-varying intensity. Epidemiological studies thus attempt to collect information on exposure values at different time points during lifetime. One of the main methodological challenges of the data analysis is to capture the temporal variation of these exposures and to assess its relationship with the risk of developing a chronic disease [1–4]. While the association between lung cancer and smoking or occupational exposure to asbestos have been extensively investigated in the literature, few studies have specifically investigated the dynamic aspects of exposure intensity over lifetime. In a recent study, we found that recent intensity of smoking had a stronger contribution than distant intensity of smoking on the risk of lung cancer and the opposite for asbestos [5], using a weighted cumulative index of exposure [6]. While such an approach allows the comparison of the risk of lung cancer associated with different hypothetical profiles of exposure over lifetime [7], it does not allow the identification of different types of longitudinal profiles of exposure intensities in the data.

Several statistical methods such as the latent class growth analysis (LCGA) [8, 9] or the latent class mixed model (LCMM) [10, 11] have been developed to identify classes of individual longitudinal trajectories of quantitative indicators. These models have been largely used to describe heterogeneous evolution of a quantitative biomarker over time [12–14], psychometric tests [15] or delinquency behavior [16] but rarely for identifying longitudinal profiles of environmental or occupational exposures. They have been used to identify classes of individual trajectories of smoking [17–19] but, to our knowledge, never to explore their association with the risk of lung cancer. Yet, such an approach could give new insight on the dose-time-response relationship between smoking or occupational exposure to asbestos and the risk of lung cancer. Indeed, identifying from the data the different profiles of exposures trajectories over lifetime and their association with the risk of cancer, without imposing any prior assumption on the number of profiles, the form of the trajectories, and the critical time-windows of exposure, may offer an alternative to commonly used dose-time-response analytical approaches including pre-specified time-windows approaches.

The objective of the present study was to evaluate to what extent classification of lifetime trajectories and comparison of their risk of cancer could give new insights on dose-time-relationships. More specifically, we used a LCMM to identify profiles of lifetime trajectories of smoking and occupational exposure to asbestos in males, and we quantified their association with the risk of lung cancer, using data from a multicentric population-based case-control study.

## Methods

The ICARE case-control study was approved by the Institutional Review Board of the French National Institute of Health and Medical Research (IRB-Inserm, n° 01–036), and by the French Data Protection Authority (CNIL n° 90120). Each subject gave a written and informed consent.

### Study design

ICARE is a large French case-control study of respiratory cancers [20]. Briefly, all histologically confirmed primary malignant lung or upper aerodigestive tract incident cancer cases, aged 18–75 years and living in 10 French geographical areas *("départements")* were recruited from French cancer registries in 2001–2007. Controls were randomly selected within the general population of the same geographical areas as the cases, using a random digit dialing procedure. Controls recruitment waves were conducted approximately every two months in 2001–2007,

and were performed in such a way that controls had a distribution by sex and age similar to that of cases, and a distribution by socioeconomic status similar to that of the general population [20].

## Data collection

Subjects were face-to-face interviewed by trained interviewers with a detailed standardized questionnaire to collect information on sociodemographic characteristics, lifetime occupational history (all jobs held for at least one month) and lifetime smoking history. For each smoking episode, start and end dates, as well as average smoking intensity (number of cigarettes smoked per day) were reported. For each job, start and end dates, industrial activity (further coded using the French Nomenclature Activities (NAF) 1999) and occupation (further coded using the International Standard Classification of Occupations (ISCO) 1968) were recorded.

## Occupational asbestos exposure assessment

Occupational exposure to asbestos was assessed by a job exposure matrix (JEM) which accounted for asbestos exposure levels changes in France over calendar time [7, 21]. For each time period and job defined as a combination of an ISCO and NAF code, the JEM assigned 1) the probability of exposure, defined as the proportion of exposed workers for that job (from 0 to 0.85, Table S1, Supplementary materials); 2) the frequency of exposure, defined as the proportion of exposed working time on a typical 8h working day for that job (from 0.025 to 0.85); and 3) the intensity of exposure, defined as the equivalent average daily concentration of asbestos fibers in the air at workplace for that job (from 0.0005 to 20 equivalent fibers per mL (f/mL)). For each subject, the level of exposure for each job held within a given calendar year was the product of intensity, probability, and frequency, and was thus expressed in equivalent f/mL. If a subject occupied several jobs over a calendar year, the mean level of exposure in that year was the average level of exposure of all jobs in that year. For example, if a subject occupied in a given calendar year two jobs, a job A with a level of exposure at 0.1 equivalent f/mL, and a job B with a level of exposure at 1 equivalent f/mL, then his mean level of exposure during that year was (0.1 + 1)/2 = 0,55 equivalent f/mL. The cumulative index of exposure (CIE) to asbestos was the sum of these annual levels over the entire occupational history, and thus expressed in f/mL-years.

## Statistical analysis

All the analyses were restricted to males who had complete information on both smoking and occupational histories. The analysis of smoking trajectories was first restricted to all ever smokers, and then to current smokers, i.e. subjects who were still smoking at the index date. The analysis of asbestos trajectories was restricted to all subjects ever occupationally exposed to asbestos.

Classes of individual lifetime trajectories of smoking and asbestos exposure were identified and compared using two separate latent class mixed models (LCMM) [11, 12, 22]. The LCMM assumes that the population is constituted of G underlying non-observed subgroups of subjects, called latent classes, with different profiles of exposure trajectory. The LCMM was made of two sub-models. Sub-model 1 (Equation 1, Supplementary materials page 5) was a multinomial logistic regression model estimating for each subject his probability to belong to each latent class. It included class-specific intercepts only.

Sub-model 2 (Equation 2, Supplementary materials page 5) was a class-specific mixed model that estimated the mean lifetime trajectory of exposure intensity in each class. The time

axis was the time before the index date (diagnosis for case, interview for control), and the time unit was a year. Smoking intensity in a given year was the number of cigarettes smoked per day (cig/day) on average in that year. Intensity of occupational exposure to asbestos in a given year was the mean level of exposure in that year (in equivalent f/mL).

To account for non-normal distribution of both smoking and occupational asbestos annual intensities, we had to use a transformation of annual exposure intensities in Sub-model 2. Because the distributions were highly skewed for both smoking and asbestos (Fig S1a and S1b, Supplementary materials), the log(x+1) transformation commonly used for the cumulative dose did not allow convergence of the estimates when applied to annual intensities. We thus used a more flexible normalizing transformation that allows normalizing distributions that are far from normal without relying on strong prior assumptions. More specifically, we used a I-splines function implemented in the lcmm R package [22, 23] with a combination of knots that best fitted the data according to Akaike Information Criteria (AIC). For smoking, we used three knots at 0, 20, 100 cig/day. For asbestos, because of the peak of annual intensities at zero (Fig S1b, Supplementary materials), no transformation function allowed the LCMM to converge when we included all subjects who were ever exposed to asbestos. We thus created an a priori class of subjects with a very low cumulative exposure over lifetime, and identified other classes using the LCMM on the remaining subjects. More specifically, the subjects who had a CIE lower than 0.26 f/mL-years (Fig S1c, Supplementary materials) were a priori classified in the class with the very low cumulative exposure. This arbitrary cut-off corresponds to the limit value of occupational exposure on 8h working day fixed by the French law (0.01 f/mL) multiplied by the observed mean duration of exposure in controls (26 years). Because the distribution of annual intensities in the remaining more substantially exposed subjects was still very skewed (Fig S1d, Supplementary materials), we used an I-spline transformation of annual intensities, with three knots at 0, 0.05, 12.6 equivalent f/mL.

To account for the abrupt changes in observed individual trajectories of intensity for both smoking and asbestos (Fig S2, Supplementary materials), we used a flexible modelling of time in Sub-model 2. More specifically, we used natural cubic splines with three inner knots, placed at quartiles of the distribution of time before the index date (12, 24, 36 years). Individual departures from the mean trajectory in each class were modelled via a random intercept with a class-specific variance. For each subject, the random intercept applied only on his entire time window of exposure (Equation 2, Supplementary materials page 5). Note that inclusion of individual random effects is an important feature of the LCMM approach since they allow the correlation between the different exposure values of the same subject to be accounted for, as opposed to the LCGA [8, 9] which does not include random effects and thus ignores this correlation.

To select the optimal number of latent classes, LCMM with different number of classes (one to six) were estimated and compared in terms of i) quality of adjustment (Bayesian Information Criteria (BIC), AIC, plot of residuals), ii) the relevance of identified trajectories, and iii) the discrimination capacity of the model based on entropy (i.e. discriminatory power, the closer to one the better) and the posterior probability classification table (Tables S3 and S4, Supplementary materials). For each estimated model, a grid of 50 initial values were tested to prevent any convergence toward a local maximum, as recommended in the GRoLTS-Checklist [24].

Once the latent classes of exposure trajectory were identified, we estimated the association between class membership and the risk of lung cancer, using weighted logistic regression. Classes of exposure trajectory were represented by dummies. Never exposed subjects were included in a reference class. To account for uncertainty of the classification resulting from the LCMM, each subject contributed to each identified latent class with a weight equal to his

posterior probability of belonging to the class. We adjusted for matching factors (age and *départements*), and considered three strategies for mutual adjustment between smoking and asbestos: (i) no mutual adjustment, (ii) adjustment for the cumulative dose of the other exposure at the index date, and (iii) adjustment for the class of trajectory of the other exposure. For (ii), we used the CIE to adjust for asbestos, and the Comprehensive Smoking Index (CSI) to adjust for smoking [25]. The CSI equals zero for never smokers, increases with average smoking intensity over lifetime and total duration of smoking, and decreases when time since smoking cessation increases to reach a minimum of zero after a very long smoking cessation [25]. The CSI also depends on a half-life and a lag parameter that have already been estimated in ICARE [26]. To account for potential nonlinear effects of age, CSI, and CIE, we used fractional polynomials. The three strategies of mutual adjustment for smoking and asbestos were compared in terms of AIC.

All the analyses were performed using the R software version 3.5, with the lcmm R package version 1.7.8 [23]. The equations of the models and the R code used to obtain the results may be found in Supplementary materials.

## Results

ICARE included 2276 male lung cancer cases and 2780 male controls. A total of 250 cases and 170 controls were excluded from the analysis because of missing information on both smoking and occupational history (Fig 1). They had similar socio-demographic characteristics as the subjects who had complete information and retained for the analysis (Table S2, Supplementary materials). Among the 2026 cases and 2610 controls retained for the analysis, 1969 (97.2%) cases and 1838 (70.4%) controls ever smoked and were used to identify profiles of smoking trajectories. For asbestos, 1417 (69.9%) cases and 1520 (58.2%) controls were ever occupationally exposed to asbestos. Among them, 505 cases and 722 controls had a CIE to asbestos lower than 0.26 f/mL-years, and were a priori classified in a class of very low cumulative exposure over lifetime. The remaining 912 cases and 798 controls were used to identify trajectory profiles of more substantial exposure to asbestos (Fig 1).

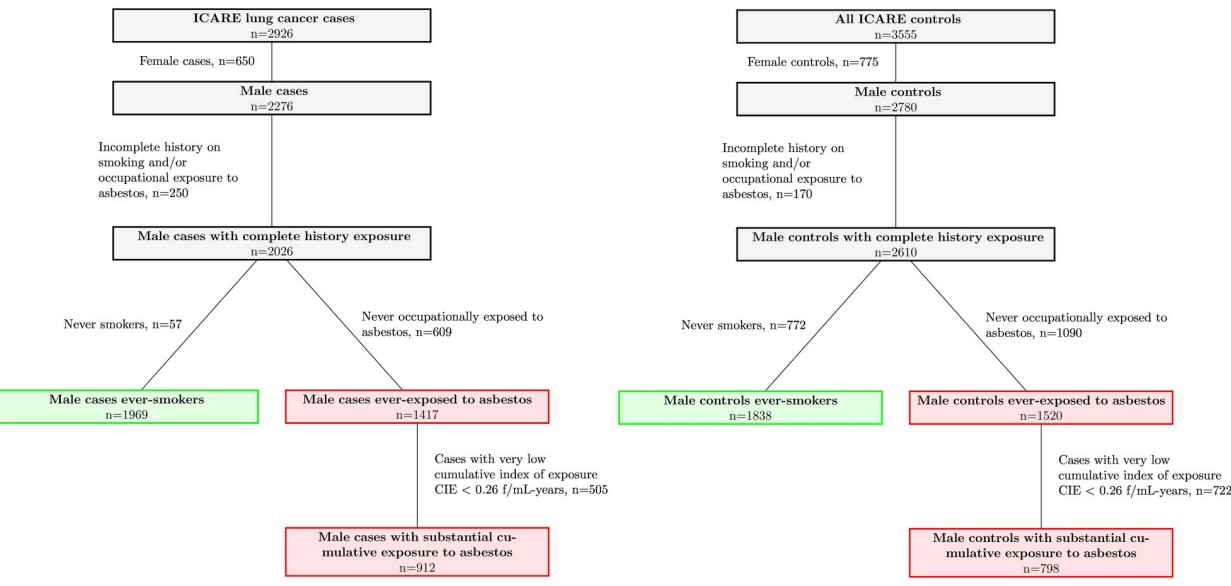

**Fig 1. Selection of subjects for the statistical analysis, ICARE case-control study, 2001–2007, France.**

**Table 1. Characteristics of cases and controls at the index date, ICARE case-control study, 2001–2007, France.**

| | Cases (n = 2026) | | Controls (n = 2610) | |
|---|---|---|---|---|
| | n (%) | Median (5th, 95th percentile) | n (%) | Median (5th, 95th percentile) |
| Age (yrs) | | 61 (45, 74) | | 59 (41, 73) |
| Area of residence | | | | |
| Calvados | 240 (11.8) | | 336 (12.9) | |
| Doubs-Belfort | 103 (5.1) | | 109 (4.1) | |
| Hérault | 227 (11.2) | | 343 (13.1) | |
| Isère | 346 (17.1) | | 375 (14.4) | |
| Loire Atlantique | 255 (12.6) | | 297 (11.4) | |
| Manche | 225 (11.1) | | 22 (8.5) | |
| Bas-Rhin | 247 (12.2) | | 331 (12.7) | |
| Haut-Rhin | 53 (2.6) | | 88 (3.4) | |
| Somme | 224 (11.1) | | 365 (14.0) | |
| Vendée | 106 (5.2) | | 144 (5.5) | |
| *Smoking exposure* | | | | |
| Status | | | | |
| Never smoker | 57 (2.8) | | 772 (29.6) | |
| Ever smoker | 1969 (97.2) | | 1838 (70.4) | |
| Current smoker | 363 (18.4) | | 203 (11.0) | |
| Ex-smoker | 1606 (81.6) | | 1635 (89.0) | |
| Duration (yrs) | 1969 | 39 (18, 54) | 1838 | 26 (4, 48) |
| Average intensity (cig/day) | 1969 | 20 (8, 40) | 1838 | 15 (1, 33) |
| Cigarette-years | 1969 | 772 (195, 1752) | 1838 | 330 (10, 1047) |
| Age at initiation (yrs) | 1969 | 17 (13, 22) | 1838 | 17 (13, 24) |
| Years since initiation | 1969 | 44 (28, 57) | 1838 | 41 (23, 55) |
| Years since cessation | 1606 | 3 (1, 29) | 1635 | 16 (1, 40) |
| CSI | 1969 | 1.8 (0.5, 2.5) | 1838 | 0.9 (0.05, 2.0) |
| *Occupational exposure to asbestos* | | | | |
| Status | | | | |
| Never exposed | 609 (30.1) | | 1090 (41.8) | |
| Ever exposed | 1417 (69.9) | | 1520 (58.2) | |
| Currently exposed | 213 (15.0) | | 226 (14.9) | |
| Ex-exposed | 1204 (85.0) | | 1294 (85.1) | |
| Duration (yrs) | 1417 | 31 (3–46) | 1520 | 26 (3–46) |
| Average intensity (f/mL) | 1417 | 0.04 (0.00001–1.2) | 1520 | 0.02 (0.0001–1.0) |
| CIE (f/mL-years) | 1417 | 1.0 (0.0002–41.6) | 1520 | 0.33 (0.0001–27.9) |
| Age at first exposure (yrs) | 1417 | 16 (14–30) | 1520 | 17 (14–31) |
| Years since first exposure | 1417 | 42 (23–58) | 1520 | 41 (18–57) |
| Years since last exposure | 1204 | 14 (1–46) | 1294 | 14 (1–46) |

CSI, Comprehensive smoking index; CIE, Cumulative index of exposure

For smoking, cases had as expected a stronger proportion of ever and current smokers at the index date (Table 1), smoked for a longer duration (median of 39 versus 26 years), stopped smoking more recently (median of 3 versus 16 years), and had a stronger average intensity over lifetime (median of 20 versus 15 cig/day) (Table 1). The best LCMM model for smoking had four latent classes (Fig 2) and had a strong discrimination capacity (Entropy

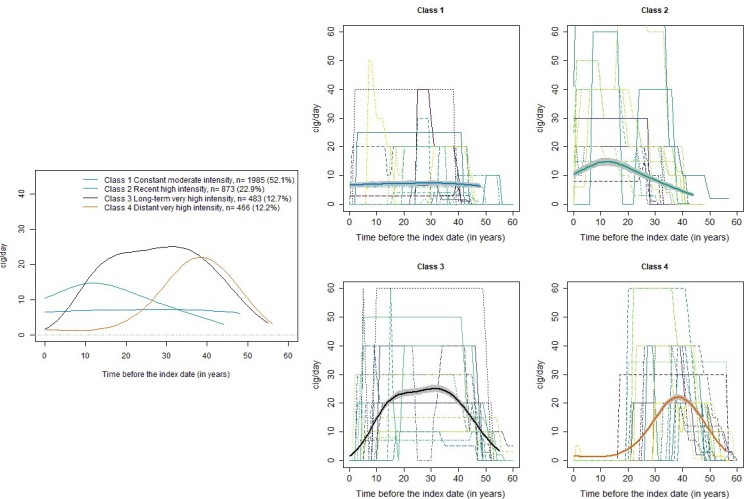

**Fig 2. Lifetime trajectories of smoking intensities, ICARE case-control study, 2001–2007, France.** The left panel shows the estimated mean trajectory of smoking intensity in the four latent classes. The right panel shows for each class, 20 randomly selected observed individual trajectories of subjects who had a high probability (close to 1) to belong to the class, with the bold line representing the estimated mean trajectory in the Class, with its 95% CI. The representation of each estimated mean trajectory is truncated at the 95th percentile of the distribution of observed exposure times in subjects a posteriori classified in the class.

of 0.97, Table S3, Supplementary materials). Class 1 (52.1% of subjects) had a constant mean trajectory of smoking intensity over lifetime, at about 8 cig/day (reference Class): "constant moderate intensity". The average trajectory of smoking intensity in Class 2 (22.9% of subjects) reached 15 cig/day within the 20 years just before the index date: "recent high intensity". Class 3 (12.7% of subjects) had very high intensities at about 25 cig/day on average, from 10 to 40 years before the index date: "long-term very high intensity". Class 4 (12.2% of subjects) had a very high-intensity episode at about 22 cig/day on average, from 30 to 50 years before index date: "distant very high intensity".

When estimating the association between the classes of smoking trajectories and the risk of lung cancer, adjusting for the cumulative dose of asbestos at the index date provided a better fit to data (AIC = 5233) compared with no adjustment for asbestos exposure (AIC = 5298), or adjustment for the class of asbestos trajectories (AIC = 5235) (Table 2). However, it did not affect the estimated association between the classes of smoking trajectories and the risk of lung cancer. As expected, the four classes of smoking trajectories had a much stronger risk of lung cancer than never smokers. Class 2 with recent high intensity of smoking had the strongest risk (OR = 41.7, 95% CI: 30.2, 57.5, compared to never smokers, Table 2). Despite men a posteriori classified in Class 2 were younger than men in all other classes (median age of 53 years at the index date), they were strong smokers (median of 800 cigarette-years) with long duration (median of 37 years). They had the lowest proportion of ex-smokers (51.8%) and the shortest time since smoking cessation (median of 1 year). Class 3 with long term very high intensity had the second strongest risk of lung cancer compared to never smokers (OR = 22.7, 95% CI: 16.0, 31.7). Class 3 thus did not have the strongest risk of lung cancer, despite it had the highest median total number of cigarette-years over lifetime (910) and the longest median duration of smoking (46 years). This is likely because the high intensities in Class 3 were mainly accumulated on average more than 15 years before the index date (Fig 2), and that compared to Class 2, it included a much higher proportion of ex-smokers (94.4% versus 51.8%) who stopped smoking for a longer duration (median of 6 years versus 1 year). Despite Class 4 had a higher

**Table 2. Association between trajectories of smoking intensity and lung cancer, ICARE case-control study, 2001–2007, France.**

| Trajectory of smoking exposure | Number of cases and controls[a] | Age at index date (years) | Cigarettes-years[b] | Smoking duration (years)[c] | Average smoking intensity (cig/day)[d] | Ex-smokers | Time since smoking cessation (years)[e] | Age at initiation (years) | OR[f] (95%CI) | OR[g] (95%CI) | OR[h] (95%CI) |
|---|---|---|---|---|---|---|---|---|---|---|---|
| | | Median (5th, 95th percentile) | Median (5th, 95th percentile) | Median (5th, 95th percentile) | Median (5th, 95th percentile) | N (%) | Median (5th, 95th percentile) | Median (5th-95th percentile) | | | |
| Never smokers | 57 772 | 59 (41, 73) | | | | | | | 1.00 | 1.00 | 1.00 |
| Class 1 Constant moderate intensity | 804 1181 | 58 (42, 72) | 350 (10, 1049) | 27 (4, 51) | 15 (1, 31) | 1871 (94.2) | 10 (1, 40) | 18 (14, 25) | 10.4 (7.8, 13.9) | 10.0 (7.5, 13.5) | 10.2 (7.6, 13.6) |
| Class 2 Recent high intensity | 612 261 | 53 (41, 66) | 800 (278, 1804) | 37 (23, 49) | 21 (9, 45) | 452 (51.8) | 1 (1, 9) | 16 (12, 22) | 44.0 (31.9, 60.6) | 41.7 (30.2, 57.5) | 42.0 (30.5,58.0) |
| Class 3 Long-term very high intensity | 337 146 | 68 (61–75) | 910 (390, 1945) | 46 (33, 55) | 20 (10, 40) | 456 (94.4) | 6 (1, 16) | 16 (12, 21) | 23.1 (16.4, 32.3) | 22.7 (16.0, 31.7) | 22.8 (16.2, 32.1) |
| Class 4 Distant very high intensity | 216 251 | 70 (62, 75) | 512 (215, 1198) | 28 (16, 45) | 19 (9, 40) | 463 (99.3) | 25 (13, 36) | 17 (13, 21) | 8.6 (6.1, 12.0) | 8.3 (5.9, 11.6) | 8.5 (6.1, 11.9) |
| AIC | | | | | | | | | 5298 | 5233 | 5235 |

OR: odds ratio; CI: confidence interval

[a] From a posteriori classification for Classes 1 to 4

[b] Sum of all annual intensities

[c] Total effective duration of smoking over all periods of smoking, excluding periods of interruptions

[d] Average intensity over all periods of smoking.

[e] Derived in ex-smokers at the index date

[f] Adjusted for age at the index date in years (second-degree fractional polynomial with powers (-2,-2)) and area of residence (*département*).

[g] Adjusted for age at the index date in years (second-degree fractional polynomial with powers (-2,-2)), area of residence (*département*), and cumulative index of occupational exposure to asbestos in f/mL-years (first-degree fractional polynomial with power 0).

[h] Adjusted for age at the index date in years (second-degree fractional polynomial with powers (-2,-2)), area of residence (*département*), and asbestos exposure trajectory class membership. This model is the same as the model in the last column of Table 3.

median total number of cigarette-years compared to Class 1 (512 vs 350), it had a moderately lower risk of lung cancer (OR = 8.3, 95% CI: 5.9, 11.6 versus OR = 10.1, 95% CI: 7.5, 13.5 compared to never smokers), likely because it had a higher proportion of ex-smokers (99.3% versus 94.2%) who stopped smoking for a much longer duration (median of 25 vs 10 years).

The results of the analysis restricted to current smokers are shown in Supplementary materials. The LCMM identified four classes of smoking trajectories in current smokers which differed in terms of smoking duration and intensity (Fig S3 in S1 File). All classes of current smokers had higher risk of lung cancer than never and ex-smokers (Table S5 in S1 File). The two classes of current smokers with strong recent intensities and shorter duration had a stronger risk of lung cancer than the two classes with lower intensity and longer duration (Table S5 in S1 File), suggesting a stronger impact of intensity than duration.

For occupational exposure to asbestos, cases had a higher proportion of ever exposed than controls (69.9% versus 58.2%, Table 1), had a longer total duration of exposure (median of 31 versus 26 years, Table 1), a stronger average intensity over lifetime (median of 0.04 versus 0.02

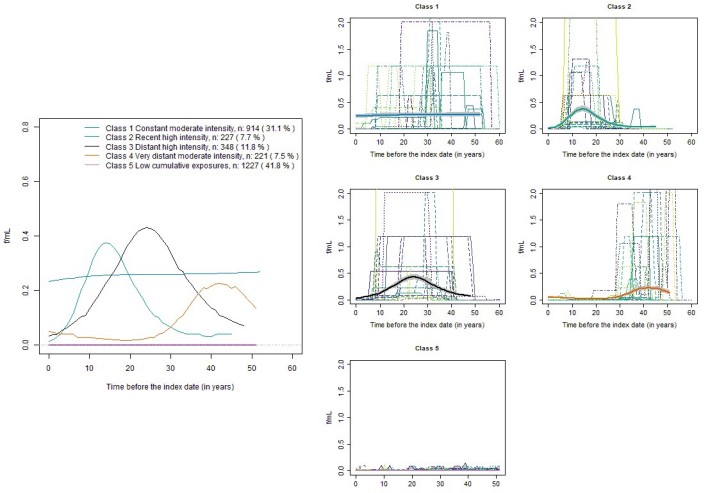

**Fig 3. Lifetime trajectories of intensities of occupational exposure to asbestos, ICARE case-control study, 2001–2007, France.** The left panel shows the estimated mean trajectory of asbestos intensity in the class of low cumulative exposure to asbestos (Class 5 including all subjects with a CIE lower than 0.26 f/mL-years) and in the four latent classes identified with the LCMM on the remaining subjects ever exposed to asbestos (Classes 1 to 4). The right panel shows for each class, 20 randomly selected observed individual trajectories of subjects who had a high probability (close to 1) to belong to the class, with the bold line representing the estimated mean trajectory in the Class, with its 95% CI. The representation of each estimated mean trajectory is truncated at the 95th percentile of the distribution of observed exposure times in subjects a posteriori classified in the class.

equivalent f/mL), and thus a stronger CIE (median of 1 versus 0.33 equivalent f/mL-years). They had similar median time since first exposure, age at first exposure, or time since last exposure (Table 1). Among subjects who had a CIE higher than 0.26 f/mL-years, the best LCMM identified four latent classes (Fig 3) and had a strong discrimination capacity (Entropy of 0.95, Table S4, Supplementary materials). Class 1 (31.1% of subjects) had a constant mean trajectory of exposure intensity at about 0.2 f/mL on average each year of occupational history: "constant moderate intensity". Class 2 (7.7% of subjects) had his highest intensity episode which reached 0.4 f/mL on average from 10 to 20 years before the index date: "recent high intensity". Class 3 (11.8% of subjects) had the highest intensity episode which reached more than 0.4 f/mL on average from 15 to 35 years before the index date: "distant high intensity". Class 4 (7.5% of subjects) had his highest intensity episode that reached about 0.2 f/mL on average from 30 to 50 years before index date: "very distant moderate intensity". All subjects who had a CIE lower than 0.26 f/mL-years were a priori classified in Class 5, which as expected had an average trajectory close to zero (Fig 3).

As expected, the classes of asbestos exposures trajectories had weaker association with lung cancer than the classes of smoking trajectories (Table 3 versus Table 2). Adjusting for the CSI provided a much better fit to data (AIC = 4454) compared with no adjustment for smoking (AIC = 6182), or adjustment for the class of smoking trajectory (AIC = 5235) (Table 3). However, whatever the adjustment for smoking, all the classes of exposed subjects, including those with a low cumulative exposure over lifetime (Class 5) had a stronger risk of lung cancer than subjects never exposed to asbestos. All the classes of more substantially exposed subjects (Classes 1 to 4) had a stronger risk of lung cancer than Class 5. The risk of lung cancer did not substantially differ between Classes 1 to 4, with all OR close to 2 (95% CI: 1.3, 2.6) when compared to never exposed subjects, and when adjusted for the classes of smoking trajectories. When adjusted for the CSI, the OR tended to moderately differ between Classes 1 to 4, with the

**Table 3. Association between trajectories of occupational exposure to asbestos intensity and lung cancer, ICARE case-control study, 2001–2007, France.**

| Trajectory of occupational exposure to asbestos | Number of cases and controls[a] | Age at index date (years) Median (5th-95th percentile) | Cumulative index of exposure (f/mL-years)[b] Median (5th-95th percentile) | Exposure duration (years)[c] Median (5th-95th percentile) | Average Intensity of exposure (f/mL)[d] Median (5th-95th percentile) | Ex-exposed N (%) | Time since last exposure (years)[e] Median (5th-95th percentile) | Time since first exposure (years) Median (5th-95th percentile) | OR[f] (95% CI) | OR[g] (95% CI) | OR[h] (95% CI) |
|---|---|---|---|---|---|---|---|---|---|---|---|
| Never exposed | 609 1090 | 60 (42, 73) | | | | | | | 1.00 | 1.00 | 1.00 |
| Class 1 Constant moderate intensity | 489 425 | 62 (43, 74) | 2.5 (0.3, 40.7) | 24 (4, 46) | 0.17 (0.01, 1.30) | 860 (94.1) | 17 (4, 46) | 44 (23, 58) | 2.0 (1.7, 2.3) | 1.7 (1.4, 2.1) | 1.8 (1.5, 2.2) |
| Class 2 Recent high intensity | 116 111 | 52 (40, 66) | 6.7 (0.4, 44.3) | 31 (18, 45) | 0.22 (0.01, 1.33) | 113 (49.8) | 5 (1, 12) | 18 (32, 51) | 2.3 (1.7, 3.0) | 1.5 (1.1, 2.1) | 1.8 (1.3, 2.5) |
| Class 3 Distant high intensity | 183 165 | 58 (45, 70) | 8.3 (0.5, 73.5) | 40 (27, 47) | 0.23 (0.01, 1.67) | 234 (67.2) | 5 (1, 18) | 42 (29, 55) | 2.0 (1.6, 2.6) | 1.7 (1.3, 2.2) | 1.9 (1.5, 2.5) |
| Class 4 Very distant moderate intensity | 124 97 | 61 (49, 72) | 6.2 (0.4, 29.9) | 39 (22, 47) | 0.17 (0.01, 0.89) | 195 (88.2) | 9 (2, 21) | 44 (33, 57) | 2.1 (1.6, 2.8) | 1.9 (1.4, 2.8) | 1.9 (1.4, 2.6) |
| Class 5 Low exposed | 505 722 | 60 (42, 73) | 0.013 (0.00006, 0.21) | 17 (2, 45) | 0.0013 (0.000006, 0.022) | 1096 (89.3) | 20 (1, 48) | 40 (15, 57) | 1.3 (1.1, 1.5) | 1.3 (1.1, 1.6) | 1.3 (1.1, 1.6) |
| AIC | | | | | | | | | 6182 | 4454 | 5235 |

OR: odds ratio; CI: confidence interval

[a] From a posteriori classification for Classes 1 to 4

[b] Sum of all annual intensities

[c] Total effective duration of smoking over all periods of exposure, excluding periods of interruptions

[d] Average intensity over all periods of exposure.

[e] Derived in ex-exposed at the index date

[f] Adjusted for age at the index date in years (first-degree fractional polynomial with power (-1)) and area of residence (*département*).

[g] Adjusted for age at the index date in years (second-degree fractional polynomial with powers (-2,-2)), area of residence (*département*), and comprehensive smoking index (second-degree fractional polynomial with powers (0.5, 3)).

[h] Adjusted for age at the index date in years (second-degree fractional polynomial with powers (-2,-2)), area of residence (*département*), and smoking trajectory class membership. This model is the same as the model in last column of Table 2.

largest discrepancy between Class 4 with the most distant asbestos exposure (OR = 1.9, 95% CI: 1.4; 2.8 compared to never exposed) and Class 2 with the most recent asbestos exposure (OR = 1.5, 95% CI: 1.1, 2.1 compared to never exposed). Class 2 also had the youngest median age (52 years) and the strongest proportion of subjects in Class 2 of smoking (35.7%, Table S6, Supplementary materials), which had the most recent high smoking intensity (Fig 3) and the strongest risk of lung cancer (Table 2). This may partly explain why adjusting for the CSI that accurately accounted for time since cessation, produced a decrease of OR from 2.3 (95% CI 1.7, 3.0) before adjustment to 1.5 (95% CI: 1.1, 2.1) after adjustment (Table 3).

## Discussion

The LCMM identified four latent classes of smoking trajectories in ever-smokers, as well four latent classes of asbestos exposure trajectories in subjects with a substantial cumulative

exposure to asbestos over lifetime, all with strong discrimination capacity. As expected, the association of lung cancer with classes of smoking trajectories was much stronger than with classes of asbestos exposure trajectories. For smoking, the class with recent high intensity (Class 2) had by far the strongest risk of lung cancer. The class of heavy ex-smokers who stopped for the longest duration (Class 4) did not have a stronger risk of lung cancer than the class of constant moderate intensity (Class 1), although they had accumulated a higher number of cigarette-years over lifetime. The trajectory analysis restricted to current smokers suggested a stronger association of lung cancer with smoking intensity than with smoking duration. For lifetime occupational exposure to asbestos, the four identified classes of asbestos exposure in substantially exposed subjects had a higher risk of lung cancer than never exposed subjects or low exposed subjects. The four classes had more or less similar risk of lung cancer, depending on smoking adjustment. When adjusted for CSI rather than classes of smoking trajectories, our results suggest a moderately higher risk of lung cancer for the class with the most distant high intensity (Class 4).

All our results are consistent with our recent study where we found stronger weights of recent intensities for smoking and distant intensities for asbestos, using a different statistical approach [5]. For smoking, the important role of recent exposures has been reported in other studies [27, 28]. The decrease in the risk of lung cancer after smoking cessation is also consistent with the literature [29, 30], confirming the importance of promoting smoking cessation. Our results in current smokers also highlight the strong impact of recent intensities, and the importance of reducing it. The strongest relative impact of smoking intensity compared to smoking duration in current smokers has been found in other studies [31]. For occupational exposure to asbestos, adjusting the OR of identified classes for the CSI provided a much better fit to data than adjusting for the classes of smoking trajectories. The CSI is indeed a quantitative measure of smoking history accounting for its three most important dimensions (intensity, duration, and time since smoking cessation). It has already been shown to provide better fit to other case-control data on lung cancer when compared to other smoking metrics [25]. The results adjusted for the CSI tend to be consistent with previous studies which have suggested that distant intensities have strong contribution to the risk of lung cancer [32, 33], confirming the importance of reduction or elimination of asbestos exposure as early as possible in the professional career. However, whatever the adjusting method for smoking, the results show an increased risk of lung cancer in all men ever occupationally exposed to asbestos, whatever the dose and the timing of exposure. It yet would be of interest to confirm all our results using the same analytical approach on other case-controls studies on lung cancer. The classes that we identified are not expected to represent the classes of exposure in the general French population, or in other general population, just because of over representation of cases, as in all case-control studies. However, the association found between the classes of trajectories and the risk of lung cancer should be reproducible in other populations.

Our study has limitations. First, we used an approximation of the annual average daily intensity of exposure for both smoking and asbestos. Exposures were retrospectively assessed, even if reported in a standardized questionnaire that was face-to-face administered by trained interviewers. For smoking, it may be reasonable to consider that the reported mean number of cigarettes smoked per day in each reported smoking period was approximately constant within the period. It has indeed been shown that self-reported smoking histories tend to be reliable [34, 35]. For asbestos, several studies have shown that self-reported occupational histories tend also to be valid [36]. To infer occupational asbestos exposure from reported job histories, we used a JEM which may to produce differential exposure misclassification when the outcome is associated with the exposure and when the exposure is categorized [37, 38]. All annual intensities of asbestos exposure over lifetime were derived from JEM and quantitatively modeled

using random effects and random measurement errors. The classification of exposure applied on trajectories of estimated intensities, but not as usual on the cumulative dose of exposure assuming no measurement errors at all. However, we acknowledge that further methodological studies are needed to investigate the impact of measurement errors generated from JEM on the results of LCMM, both in terms of classification and mean estimated trajectories within each class.

Another limitation of our study is that we had to exclude from the LCMM analysis for asbestos, subjects who had accumulated a very low dose of exposure over their entire occupational history. Indeed, while the LCMM is an advanced statistical method that accounts for strong correlation between exposure intensities of the same subject, as opposed to LCGA [8, 9], its current version does not handle very high proportions of zeros in exposure trajectories. We used an I-spline transformation of intensities, but it was not sufficient to solve convergence issues when including all subjects. Further methodological developments are thus needed to better handle highly skewed distributions of exposures with pike at zero due to non-exposed periods, in the LCMM. However, until such extensions are developed, a solution is to a priori groupvery low exposed subjects in a specific class, describe their mean exposure trajectory and characteristics, and compare their risk of disease to other identified classes of exposure trajectory as well as to never exposed, as we did in the present analysis.

In conclusion, our study provides an illustration of the dynamics of smoking and occupational exposure to asbestos and their association with the risk of lung cancer. From a more general point of view, we believe that LCMM opens new perspectives for the analyses of dose-time-response relationships between protracted exposures and the risk of developing a chronic disease like cancer, by providing a complete picture of exposure history in terms of intensity, duration, and timing of exposure.

## Supporting information

**S1 File.**
(PDF)

## Author Contributions

**Conceptualization:** Karen Leffondré.

**Data curation:** Danièle Luce, Pascal Guénel, Isabelle Stücker.

**Formal analysis:** Emilie Lévêque.

**Funding acquisition:** Aude Lacourt, Karen Leffondré.

**Methodology:** Cécile Proust-Lima, Karen Leffondré.

**Software:** Emilie Lévêque, Viviane Philipps.

**Supervision:** Karen Leffondré.

**Validation:** Aude Lacourt, Danièle Luce, Pascal Guénel, Isabelle Stücker, Cécile Proust-Lima.

**Visualization:** Emilie Lévêque.

**Writing – original draft:** Emilie Lévêque.

**Writing – review & editing:** Emilie Lévêque, Aude Lacourt, Danièle Luce, Pascal Guénel, Cécile Proust-Lima, Karen Leffondré.

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
