## [Decision Letter · Decision Letter 0]

1 Apr 2020

PONE-D-20-04889

A new approach for investigating the association between an environmental or occupational exposure over lifetime and the risk of cancer:

Application to smoking, asbestos, and lung cancer.

PLOS ONE

Dear Dr Leffondré,

Thank you for submitting your manuscript to PLOS ONE. After careful consideration, we feel that it has merit but does not fully meet PLOS ONE’s publication criteria as it currently stands. Therefore, we invite you to submit a revised version of the manuscript that addresses the points raised during the review process.

We would appreciate receiving your revised manuscript by May 16 2020 11:59PM. To enhance the reproducibility of your results, we recommend that if applicable you deposit your laboratory protocols in protocols.io, where a protocol can be assigned its own identifier (DOI) such that it can be cited independently in the future. For instructions see: http://journals.plos.org/plosone/s/submission-guidelines#loc-laboratory-protocols

We look forward to receiving your revised manuscript.

Kind regards,

Raymond Niaura, PhD

Academic Editor

PLOS ONE

Journal Requirements:

1. We note that you have indicated that data from this study are available upon request. PLOS only allows data to be available upon request if there are legal or ethical restrictions on sharing data publicly. For information on unacceptable data access restrictions, please see http://journals.plos.org/plosone/s/data-availability#loc-unacceptable-data-access-restrictions.

Reviewers' comments:

Reviewer's Responses to Questions

**Comments to the Author**

1. Is the manuscript technically sound, and do the data support the conclusions?

Reviewer #1: Partly

2. Has the statistical analysis been performed appropriately and rigorously? 

Reviewer #1: Yes

3. Have the authors made all data underlying the findings in their manuscript fully available?

Reviewer #1: No

4. Is the manuscript presented in an intelligible fashion and written in standard English?

Reviewer #1: Yes

5. Review Comments to the Author

Reviewer #1: This reviewer believes that authors undertook important effort to utilize statistical tools that are unjustly under-used in occupational epidemiology. Their approach should help derive more information from available data by getting away from bias that can arise from pre-specified time windows of exposure. It must be noted that both approaches have merit, especially if pre-specified time windows are informed by a priori hypotheses, thereby reducing bias that can arise in data- (rather than hypothesis-) driven analysis. This reviewer hopes that the authors see the merit of this critique of trajectory analyses and are willing to offer their opinion on this matter the introduction to the paper. But more importantly, this reader believes that the authors under-stated the importance of some of their etiological findings, which were enabled by novel method of analysis. (The authors discuss their etiological findings in light of prior research and note congruence, but it seems justified to be more assertive about the value of author’s research to advancing policy-relevant knowledge.) Thus, this reviewer aimed his comments at strengthening the impact of the manuscript and anticipating challenges to validity of some of its more important conclusions to public health. See attached for details.

6. PLOS authors have the option to publish the peer review history of their article (what does this mean?). If published, this will include your full peer review and any attached files.

Reviewer #1: Yes: Igor Burstyn

---

## [Author Response · Author response to Decision Letter 0]

30 Jun 2020

We thank the reviewer for careful and in-depth revision of our manuscript, and for very insightful and useful suggestions that helped us to significantly improve our manuscript. Please find our response to each point in the attached file name "Response_to_Reviewers".

---

## [Editor Report · Decision Letter 1]

14 Jul 2020

A new trajectory approach for investigating the association between an environmental or occupational exposure over lifetime and the risk of chronic disease: Application to smoking, asbestos, and lung cancer.

PONE-D-20-04889R1

Dear Dr. Leffondré,

We’re pleased to inform you that your manuscript has been judged scientifically suitable for publication and will be formally accepted for publication once it meets all outstanding technical requirements.

Kind regards,

Raymond Niaura, PhD

Academic Editor

PLOS ONE
---

## [Editor Report · Acceptance letter]

30 Jul 2020

PONE-D-20-04889R1 

A new trajectory approach for investigating the association between an environmental or occupational exposure over lifetime and the risk of chronic disease: Application to smoking, asbestos, and lung cancer. 

Dear Dr. Leffondré:

I'm pleased to inform you that your manuscript has been deemed suitable for publication in PLOS ONE. Congratulations! Your manuscript is now with our production department. 

Kind regards, 

on behalf of

Dr. Raymond Niaura 

Academic Editor

PLOS ONE